# Current Insights into the Use of Probiotics and Fatty Acids in Alleviating Depression

**DOI:** 10.3390/microorganisms11082018

**Published:** 2023-08-05

**Authors:** Han Gao, Chengwei He, Shuzi Xin, Rongxuan Hua, Yixuan Du, Boya Wang, Fengrong Gong, Xinyi Yu, Luming Pan, Lei Gao, Jingdong Xu

**Affiliations:** 1Department of Clinical Laboratory, Aerospace Center Hospital, Beijing 100049, China; gaohan703851@163.com; 2Department of Physiology and Pathophysiology, School of Basic Medical Sciences, Capital Medical University, Beijing 100069, China; hcw_1043@163.com (C.H.); xinshuzi@gmail.com (S.X.); 3Department of Clinical Medicine, School of Basic Medical Sciences, Capital Medical University, Beijing 100069, China; andrewhdd@126.com (R.H.); duyixuan0312@163.com (Y.D.); gfr0086@163.com (F.G.); a2772520017@163.com (X.Y.); plm03262022@163.com (L.P.); 4Department of Digestive Oncology, Peking University Cancer Hospital, Beijing 100044, China; wbysonnig@163.com; 5Department of Biomedical Informatics, School of Biomedical Engineering, Capital Medical University, Beijing 100069, China; bmi5@ccmu.edu.cn

**Keywords:** probiotics, fatty acids, depression, gut microbiome, meta-analysis, RCT

## Abstract

(1) Background: Depression is the most prevalent psychiatric symptom present among individuals of all ages and backgrounds, impacting an estimated 300 million people globally. Therefore, it demands a significant amount of attention when it comes to managing depression. A growing amount of data reveal that probiotics and fatty acids could be beneficial to depression. However, the opposing position maintains that they have no influence on depression. A network meta-analyses of existing datasets aid in the estimation of comparative efficacy as well as in achieving an understanding of the relative merits of different therapies. The purpose of this study was to investigate the current evidence for probiotic or fatty acid depression therapy and to establish a practical alternative for depression patients using a meta-analysis and metagenomic data from a Wistar–Kyoto (WKY) depressed rat model. (2) Methods: Probiotic data were obtained from seven randomized controlled trial studies (*n* = 394), and fatty acid data were obtained from 24 randomized controlled trial studies (*n* = 1876). Meanwhile, a metagenomics analysis of data on animal gut flora was also applied to validate the preceding evidence. (3) Results: The fatty acid studies were separated into three sections based on the duration of probiotic delivery: ≤8 weeks, 9–12 weeks, and >12 weeks. The results were as follows: for ≤8 weeks, MD = −1.65 (95% CI: −2.96–−0.15), *p* = 0.01; for 9–12 weeks, MD = −2.22 (95% CI: −3.03–−1.22), *p* < 0.001; for >12 weeks, MD = −1.23 (95% CI: −2.85–0.39), *p* = 0.14. Regarding the probiotics, the meta-analysis revealed MD = −2.19 (95% CI: −3.38–−2.43), *p* < 0.001. The research presented herein illustrates that probiotics and fatty acids may successfully lower depression scores. Additionally, the probiotics were drastically reduced in the WKY rats. (4) Conclusions: According to the data, a depression intervention utilizing probiotics outperformed the control, implying that the use of probiotics and fatty acids may be a successful strategy for depression treatment.

## 1. Introduction

The global prevalence of depression has increased considerably since 1990, resulting in severe disability and a loss of fulfillment in life [1,2]. Depression is characterized by a persistent sense of melancholy, social withdrawal, and/or an inability to feel pleasure (anhedonia), as well as dysfunctions such as changes in the intake of food and sleep, difficulties focusing, and irritability [3]. According to a report from the World Health Organization (WHO), depression affects around 350 million people worldwide [4]. Furthermore, depression raises the risk of suicidal behavior and mortality [5], and the total number of suicides owing to depression approaches 800,000 per year (WHO, 2017). As a result, the increasing prevalence of depression presents a huge challenge for healthcare systems, emphasizing the importance of optimizing screening, diagnosis, and therapy.

In primary care settings, a score of 2 or higher on the Patient Health Questionnaire 2 (PHQ-2) survey, which is widely utilized for diagnosis, has a sensitivity of 86% and a specificity of 78% [6]. Following a positive PHQ-2 depression result, the PHQ-9 is a diagnostic and severity evaluation tool [7]. The PHQ-9 has been approved for use in primary care settings since it is straightforward to score and can evaluate the severity of MDD. The Beck Depression Inventory and the *Diagnostic and Statistical Manual of Mental Disorders, Fifth Edition* (DSM-5) [8], are two widely utilized tools. In terms of therapy, a wide range of antidepressants with somewhat distinct mechanisms of function are frequently employed and are available globally. Simultaneously, non-pharmacological therapies such as exercise, prebiotics, probiotics [9], yoga, fatty acids [10], self-help books, exercise, relaxation therapy, and acupuncture have received widespread acceptance due to substantial clinical evidence of their beneficial effects [11]. Meanwhile, several studies have identified different biomarkers linked to depression, such as folate [12], Vitamin D [13], zinc, iron [14,15], and nestin-1 [16].

Other studies, however, have raised questions about the usefulness of omega-3 fatty acids and probiotics in the treatment of depression in adults [17]. It has now become obvious that the enormous amount of gut microbiota may have a crucial influence on our mental health. Through animal models, dysbiosis has been demonstrated to impair vagus signal transduction and impede protein synthesis in the hippocampus, which might be reversed by rescuing the gut microbiome via the administration of specific strains of probiotics or by gradually recovering through the elimination of the relevant stressor(s) [18,19] and probiotics [20]. *Lacticaseibacillus rhamnosus* supplementation in rats for 28 days substantially lowered depression evaluation scores [21]. Meanwhile, 30 days of *Lactobacillus helveticus R0052* and *Bifidobacterium longum R0175* supplementation for 30 days can considerably lower psychological stress levels in rats and humans [22]. Individuals with MDD who received probiotics had lower Beck Depression Inventory scores [23]. Furthermore, an animal investigation into the mechanism of depression alleviation revealed that probiotics may boost tryptophan levels in the plasma while reducing serotonin concentrations in the frontal brain and cortical dopamine metabolites [24]. There are currently several types of research on depression, probiotics, and fatty acids. The benefits of probiotics and fatty acids on depression treatment, on the other hand, are still being debated. The purpose of this research is to review and analyze the existing evidence to determine whether probiotics and fatty acids have antidepressant effects.

## 2. Materials and Methods

### 2.1. Data Sources and Study Selection

All titles and abstracts were independently reviewed by pairs of academics, with no restrictions on language or year of publication. The databases and other sources, including PubMed, EMBASE, Cochrane Library, and Web of Science, were sensitively and thoroughly searched for studies from the earliest record to 1 July 2022, using the text keywords (title or abstract) and Medical Subject Headings (MeSH) terms “probiotics”, “prebiotics”, “*Lactobacillus*”, “*Bifidobacterium*”, and “depression” to identify all relevant studies. Except for the terms “fatty acids” and “depression,” the search approach of the researchers concerning fatty acids was consistent with the previous approach. Full texts were also reviewed for cases in which eligibility could not be established based on the title and abstract.

### 2.2. Inclusion Criteria

The following inclusion criteria were fulfilled by the studies:(1)A randomized controlled trial (RCT);(2)A clinical cohort and controls, with the clinical cohort’s intervention being the administration of probiotics or fatty acids;(3)Reports that utilized identical methodologies and scientific grading scales for depression;(4)Data reported as means ± SDs.

### 2.3. Extraction and Synthesis of Data

The estimates were pooled by standardizing the standard form, which included the first author’s name, the year of publication, the nation, the sample, the research procedure, the depression rating scales, probiotic species, and fatty acid descriptions.

### 2.4. Assessment Quality

To assess the risk of bias, Review Manager 5.4 was used, which is divided into seven domains: random order generation (selection bias), allocation concealment (selection bias), blinding of participants and personnel (performance bias), blinding of outcome assessment (detection bias), incomplete outcome data (attrition bias), selective reporting (reporting bias) and other bias.

### 2.5. Statistical Analysis

Review Manager version 5.4 was utilized to create the forest plots, funnel plots, and risk of bias map (the Nordic Cochrane Centre, Copenhagen, Denmark). Heterogeneity between studies was exhibited by standard χ^2^ tests (*p*-value < 0.10), and the *I*^2^ statistic revealed research heterogeneity [25,26]. *I*^2^ values of roughly 25% indicate low heterogeneity, *I*^2^ values of 50% indicate moderate heterogeneity, and *I*^2^ values of 75% indicate significant heterogeneity [26]. The random effects model incorporates heterogeneity from not only sampling but also study-level errors by using the mean of a range of actual effect values rather than a single true effect size. Fixed effects models, on the other hand, have a single real effect size that occurs across all trials, and any volatility observed is purely due to sampling error, resulting in estimates that only include within-study variance. Random effects models are more exact than fixed effects models in circumstances of substantial heterogeneity. To explore probable reasons for heterogeneity, the following criteria were planned: treatment duration, probiotic species or strain(s), and probiotic dose. The subgroup analysis was divided based on the duration of the intervention lasted. Sensitivity analyses were performed by eliminating each study sequentially to determine the influence of individual studies on total prevalence estimates. The random effects model was used to assess the influence of fatty acid delivery on depression ratings, and the findings were presented as the standardized mean difference (SMD) of 95% confidence intervals (CIs). The depression score for continuous variables, as well as the associated 95% CIs, was calculated using fixed effects models for probiotics therapy.

### 2.6. Metagenome Analysis

Metagenomic DNA was extracted from WKY rat feces for quantitative investigation of gut microbiota utilizing the Stool Genomic DNA Kit (GENEWIZ, Suzhou, China). To the 3’end of the purified DNA fragments, an “A” nucleotide was inserted for end-repair. Following that, palindromic forked adapters embedded with unique 8-base index sequences were attached to each end of paired-indexed Illumina dual-end adapters. To complete the development and detection of the sequencing library, target DNA fragments were screened using magnetic beads and amplified with PCR with an index at the end of the target segment. Illumina’s TruSeq ChIP Library Preparation kit was used to create sequencing libraries, and libraries were barcoded using an Illumina HiSeq2000 instrument depending on fragment size.

## 3. Results

Following selection based on the inclusion criteria, research relevant to the region was assessed to validate whether it was appropriate for inclusion in this review. Data were extracted from 24 RCT studies, which included 1876 samples for fatty acid treatment (as depicted in Figure 1), and 7 RCT studies that included 394 participants for probiotics treatment, with a primary diagnosis of depression according to standard diagnostic criteria such as Beck Depression Inventory-II (BDI-II), Montgomery-sberg Depression Rating Scale (MADRS), Patient Health Questionnaire (PHQ-9), Depression Interview, and Structured Hamilton Scale (HSCL-D-20), Geriatric Depression Scale (GDS) and Depression Scale of the Center for Epidemiological Studies(CES-D).

### 3.1. Fatty Acids Intervention

Figure 1 depicts the screening study procedure. In all, 24 studies and 1876 persons were included, with detailed information presented in Table 1. The risk of bias for the included RCTs remained low, as demonstrated in Figure 2. The depression score was lowered in the fatty acid administration vs. the placebo, with MD = −1.73 (95% CI: −3–−0.46, *I*^2^ = 65%, *p* = 0.008) (Figure 3 and Figure 4).

A subgroup analysis was carried out to identify potential sources of heterogeneity between studies and to evaluate the consistency of the findings across different patient subpopulations. However, several considerations should be taken when evaluating the reported effect sizes and their possible consequences. The study was then separated into three sections based on the duration of fatty acids treatment: ≤8 weeks, 9–12 weeks, and >12 weeks. As Figure 5 indicates, the ≤8 weeks, MD = −1.65 (95% CI: −2.96–−0.15), *p* = 0.01; 9–12 weeks, MD = −2.22 (95% CI: −3.03–−1.22), *p* < 0.001; >12 weeks, MD = −1.23 (95% CI: −2.85–0.39), *p* = 0.14. The results revealed that the lessened depression scale in the fatty acid treatment group was better than that in the placebo groups, and there was considerable statistical heterogeneity in the 9–12 weeks subgroup analysis (*I*^2^ = 71%, *p* < 0.001).

### 3.2. Probiotics Intervention

A flow diagram of probiotic-related references is presented in Figure 6 and Figure 7, and 394 participants were evaluated in total, with detailed information summarized in Table 2. According to the survey’s quality evaluation, the risk of bias for each RCT included was minor (Figure 7). The meta-analysis revealed MD = −2.19 (95% CI: −3.38–−2.43), *p* < 0.001, confirming that the probiotics intervention was more effective than the placebo in alleviating depression symptoms (Figure 8 and Figure 9).

### 3.3. Evaluation of Probiotic Content in the Rat Model of Depression

Based on the evidence presented above, we evaluated the number of depressed rats. Since the WKY rat is autonomously more susceptible to stress and represents a well-known genetic model of endogenous depression [59,60,61]. When stressed, in WKY rats, the NE system is wrongly activated in WKY rats, resulting in aberrant monoamine alterations and enhanced stress susceptibility [62,63]. As a basis, we selected the WKY rat as the depression model and the SD rat as the control sample. Furthermore, we performed metagenome analysis to assess the composition of bacteria in the feces of WKY rats and SD rats to determine whether the level of probiotics is lowered in the depression model. Figure 10 reveals that the SD rats possessed 68 unique Lactobacillus species, but the WKY rats only had nine different Lactobacillus species. Moreover, there were 28 unique species of Bifidobacterium in SD rats as well as 11 species in WKY rats. Additionally, as depicted in Figure 10, the number of commonly encountered species decreased significantly in the WKY rats as opposed to the SD rats. The foregoing findings revealed that the abundance and diversity of probiotics had dramatically decreased in WKY rats.

## 4. Discussion

Depression has attracted a great deal of attention owing to its broad occurrence. As a corollary, despite substantial study into its diagnosis, treatment, and origin, many elements remain obscure and are being explored. So far, three well-known key mechanisms of depression development have been identified: disruption of the hypothalamus–pituitary–adrenal (HPA) axis regulation [64] and the probable significance of adult-generated neuron persistence in the hippocampus’s dentate gyrus [65], as well as declination of the neurotransmission of brain monoamines. However, the causes of depression are not well understood. Numerous studies have demonstrated that there exists a great deal of potential biological correlates. Peripheral inflammation makers in both blood and cerebrospinal fluid have been linked to depression development in studies involving humans and animals [66]. Therefore, neuroinflammation has gained increasing attention and is regarded as a key role in depression; IL-6, IFN-α, and C-reactive protein (CRP) have been shown to be linked to depression [67,68]. In addition, evidence indicates that increased pro-inflammatory kinds of cytokines correlate to depression symptoms [69]. Overall, there is considerable evidence that immune system modifications may be one of the mechanisms through which antidepressant medicines work. There is significant evidence for a relationship between inflammatory activity and depression. Moreover, a considerable decline in sex hormones, such as estrogen and progesterone, may exacerbate the disorder’s development [70]. Another piece of research revealed that one of the reasons for depression is low serotonin levels or aberrant serotonin metabolism, which disturbs the serotonergic system [71,72,73]. Deficits in micronutrients such as vitamins (B complex, D) [74,75], omega-3 polyunsaturated fatty acids (PUFAs), and minerals (Zinc, magnesium) [76] can also contribute to depression [77,78,79].

Omega-6 and omega-3 PUFAs are routinely used to prevent or treat depression. Linoleic acid (C18:3n3) is normally transformed into two g-linolenic acids (C18:3n6), which are subsequently turned into arachidonic acids (AA, 20:4n-6 or 20:4ω6). Linolenic acid is converted into docosahexaenoic acid (DHA, 22:6n-3 or 22:6ω3) and eicosapentaenoic acid (EPA, 20:5n-4 or 20:5ω3). It has been shown that AA and DHA are structurally essential parts of the phospholipid membranes of synaptic terminals and the brain and play a vital role in neuronal membrane maintenance, neurotransmission, and signal transduction, as well as influencing the serotonin hormone metabolism, thereby influencing mood, sleep, and pleasure behaviors [80,81,82,83,84].

Long-term DHA deficiency, combined with a low level of 5-hydroxy indoleacetic acid in cerebrospinal fluid, could have an impact on brain structure and function [85], resulting in decreased receptor activities of serotonin release and reuptake, lowered serotonergic neurotransmission efficiency, and lessened serotonin metabolite [71,86,87]. Fatty acid binding proteins 5 and 7 [88], as well as the transporters Mfsd2a [89], are thought to modulate the absorption of blood DHA across the blood–brain barrier. Furthermore, cell integrity plays an integral part in DHA replenishment in the brain.

EPA is essential in the regulation of immunological and inflammatory responses. It should be highlighted that EPA inhibits prostaglandin E2 synthesis [90], NF-κB activation, and the production of pro-inflammatory cytokines such as TNF-α, IL-1β, IL-6, IL-8, and IFN-γ [91], thereby diminishing the response to inflammatory stimuli [92]. Pro-inflammatory cytokines are supposed to alleviate depression by increasing tryptophan conversion to kynurenic acid and subsequently hindering tryptophan conversion to serotonin. A distinct perspective was taken on chronic stress, depression, and immunity [93]. Tryptophan is a precursor of the kynurenine pathway (KP), which plays a crucial part in the development of depression [94,95]. Females are disproportionately impacted by stress-primed inflammation, which explains why women are higher vulnerability to depression than men [96]. EPA can also enhance serotonin release by inhibiting the synthesis of PGE2, which has an inhibitory influence on serotonin release from presynaptic neurons [97,98], thereby resulting in an anti-depressive impact.

A growing body of evidence suggests that gut health is linked to psychological health, implying that the gut–brain axis exists and communicates in the gastrointestinal tract, central nervous system (CNS), autonomic nervous system, enteric nervous system, neurohormonal system, and immune system in complex and bidirectional ways [99]. Notably, the gut microbiota contributes to depression by regulating various signaling pathways implicated in the gut–brain axis, such as inflammatory responses, the host metabolism, kinase II/cyclic AMP response element-binding protein, and mitogen-activated protein (MAP) kinase signaling, Ca^2+^/calmodulin-dependent protein, and the endocannabinoid system [100,101,102,103,104]. With the advancement of understanding the gut microbiota–gut–brain axis, gut microbiota may play a potentially crucial role in depression diagnosis and therapy owing to their role in the bidirectional communication between the gastrointestinal tract and the brain. A disturbed microbiome balance and functional alterations owed the characteristic of increased pro-inflammatory bacteria (e.g., *Desulfovibrio* and *Escherichia/Shigella*) and decreased anti-inflammatory butyrate-producing bacteria (e.g., Bifidobacterium and Faecalibacterium). *Candida albicans* and *Staphylococcus aureus* were shown to be dramatically raised in depressed rat models, whereas *Lactobacilli* and *Bifidobacteria* were discovered to be significantly lowered [99]. Similar to what had been discovered in rat models, *Bacteroidales* were abundant in depressed individuals, whereas the family *Lachnospiraceae* was underrepresented [99]. In China, depression was associated with higher levels of *Eggerthella* and *Acidaminococcus* but lower *Coprococcus* and *Fusicatenibacter* [105].

Restoring commensal microbiota, on the other hand, may partially reverse the psychiatric disorders. Probiotics have been shown to influence a psychological trait linked to depression vulnerability and have a suppressive effect on depression. As we all know, prebiotics and probiotics have gained popularity, owing in part to their lack of cognitive side effects and few addiction instances until lately. *Lactobacillus* and *Bifidobacterium* were the most commonly utilized probiotic supplements to relieve the depressive phenotype (Table 3). *Streptococcus*, *Bacillus*, *Akkermansia*, and *Faecalibacterium* can attenuate the depressive phenotype. However, contrary statements have also asserted that ingestion of *L. intestinalis*, *L. reuteri*, and *L. helveticus* can cause depressive- and pessimism-like phenotypes as well as disruption of social withdrawal [106,107]. Prolonged (21-day) administration of the probiotic *Bifidobacterium adolescentis*, a Gram-positive anaerobic bacterium that makes up the colon microbiota in animals [108], has antidepressant properties by lowering inflammatory cytokines and normalizing the gut microbiota [109]. There are few notable studies concerning systemic and gut inflammation. Despite the potential for prebiotics and probiotics to treat these disorders, research detected an enrichment tendency for the specific strain of *Lactobacillus* in patients with MDD [110], implying that the line between beneficial and pathogenic bacteria in depression is blurred. Restoring a balance may be more critical than regulating germs from a certain category.

It is worth considering how probiotics might help with depression symptoms. Probiotics have been demonstrated to regulate the immune system, promote antimicrobial substance synthesis, limit harmful microorganisms competitively, and strengthen the epithelial barrier and gut mucosal adhesion [111]. Additionally, probiotics may interfere with opioid and cannabinoid receptors in gut epithelial cells, as well as calcium-dependent potassium channels in gut sensory neurons [112,113]. These receptors have a function during neuroinflammatory processes caused by glial cells, including astrocytes and microglia [114]. Probiotics, similar to fatty acids, have an influence on serotonin. Bacteria such as *Bifidobacterium infantis*, *Streptococcus*, *Escherichia*, and *Enterococcus*, can elevate serotonin levels in the brain by enhancing tryptophan availability [24]. Meanwhile, probiotics alter neurotransmitters implicated in the pathophysiology of depression. Norepinephrine (NE) is produced by *Escherichia*, *Bacillus*, and *Saccharomyces*, whereas dopamine is produced by *Bacillus* and *Serratia*. The impact of certain lactic acid bacteria species was on biogenic amine production by a foodborne disease [115]. *Lactobacillus* and *Bifidobacterium* can also manufacture gamma-aminobutyric acid (GABA) [116]. Probiotics’ anti-depressive impact is aided by pro-inflammatory substances as well as neurotransmitters. Oral use of the probiotic *Bifidobacterium infantis* 35624 results in a high level of IL-10 in the peripheral blood [117]. Besides, *Lactiplantibacillus plantarum Plantarum* and *Bifidobacterium longum* also suppress NLRP3-mediated IL-1 generation in microglia cells [118]. *Lactobacillus reuteri NK33* and *Bifidobacterium adolescentis NK98* have been demonstrated to lower IL-6 and corticosterone levels in the blood [119].

In this paper, there existed some limitations that restricted the preciseness of the analysis. First, perhaps most important, the divert criterion of judging and assessing depression may partly account for the heterogeneity. The patients’ different ages were another key factor that contributed to decreased accuracy. It should also be mentioned that the antidepressant was not included in all trials. Last but not least, the therapy was inconsistent across species and time. As a result, the following research should be more detailed and limited to a more manageable group. Only in this manner can scientific and meaningful research be performed.

## 5. Conclusions

The current study presents the most thorough meta-analysis of data from published obtaining data from published research on the effects of fatty acids or probiotics on depression. At the same time, we investigated the link between intestinal flora and depression using data from the depression rat model’s metagenome. All of these findings demonstrate that fatty acids and probiotics have a beneficial effect on depression, providing a novel treatment strategy for future clinical therapy of depressed individuals.

## Figures and Tables

**Figure 1 microorganisms-11-02018-f001:**
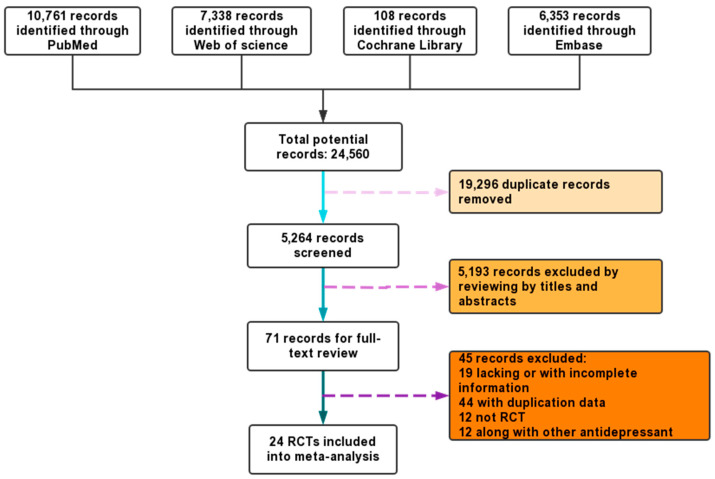
The search process and the screening of the articles for identifying the eligible studies. 24 RCTs were involved in the analysis.

**Figure 2 microorganisms-11-02018-f002:**
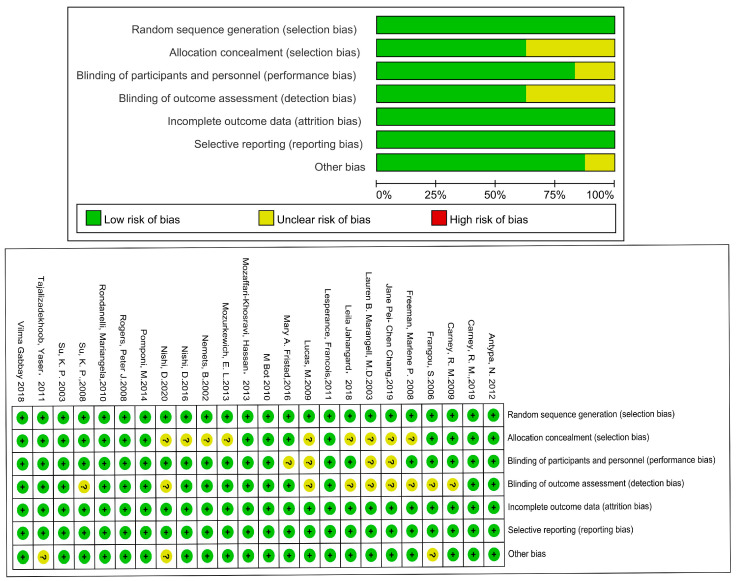
Risk of bias of included studies in fatty acid treatments using the Cochrane Collaboration’s tool. Risk of bias graph: each risk of bias item was presented as percentages in all included studies and risk of bias summary.

**Figure 3 microorganisms-11-02018-f003:**
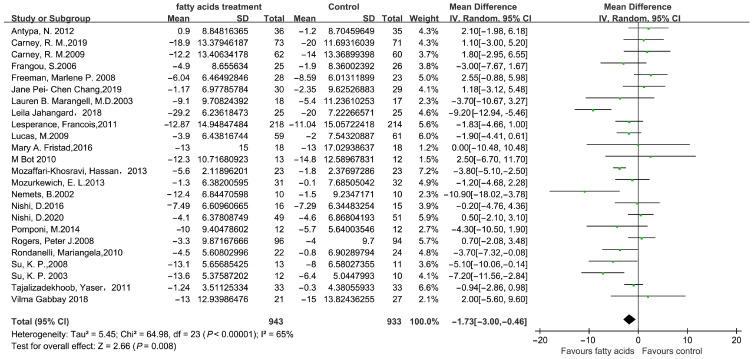
The forest plot from included studies: probiotics vs. placebo. ♦ represents the total Mean Difference. The green square represents Mean Difference and its size reveals weight. The black line represents 95% CI.

**Figure 4 microorganisms-11-02018-f004:**
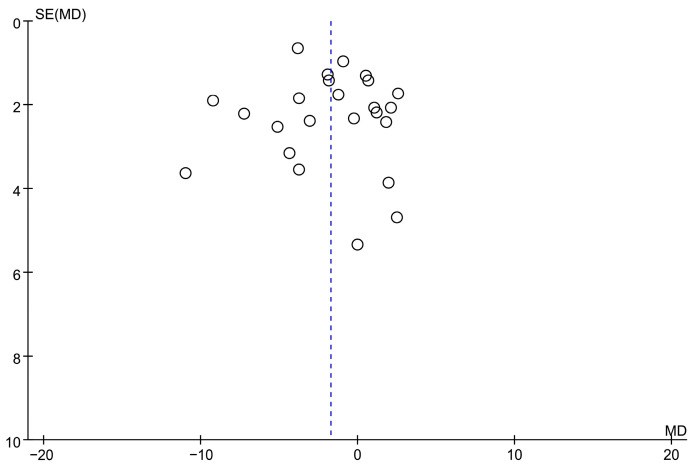
A funnel plot of included research refers to probiotic treatment.

**Figure 5 microorganisms-11-02018-f005:**
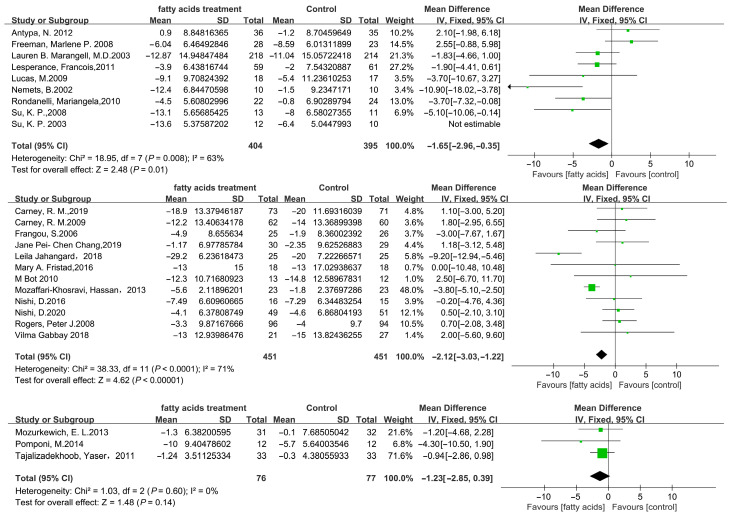
Forest plot of subgroups relating to fatty acids versus placebo: influence on depression scores. The results revealed that the lessened depression scale in the fatty acid treatment group was better than that in the placebo group. ♦ represents the total Mean Difference. The green square represents Mean Difference and its size reveals weight. The black line represents 95% CI.

**Figure 6 microorganisms-11-02018-f006:**
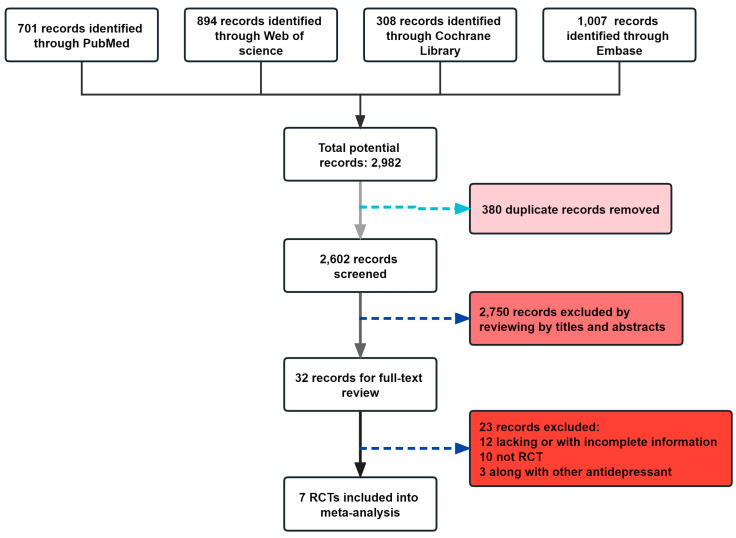
Flow diagram depicting the therapeutic search and selection technique for probiotics. 7 RCTs were included in the analysis.

**Figure 7 microorganisms-11-02018-f007:**
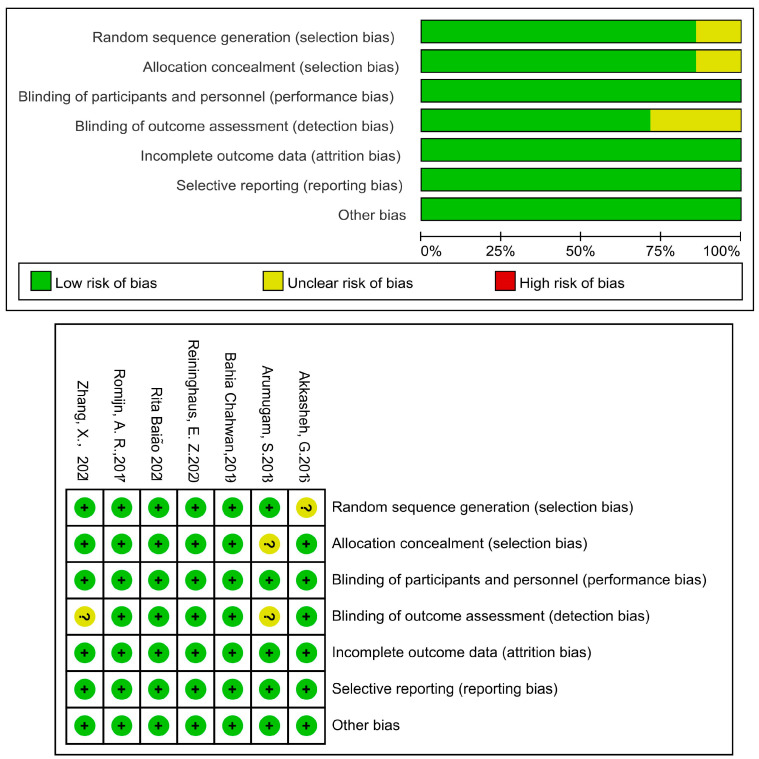
Risk of bias of illustrated studies in probiotics therapy using the Cochrane Collaboration tool. Risk of bias graph: each risk of bias item was presented as percentages in all analyzed investigations and risk of bias summary.

**Figure 8 microorganisms-11-02018-f008:**
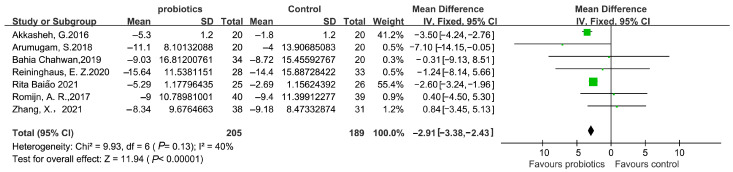
Forest plot of depression scores from included studies referring probiotics vs. placebo. The probiotics intervention was more effective than the placebo in alleviating depression symptoms. ♦ represents the total Mean Difference. The green square represents Mean Difference and its size reveals weight. The black line represents 95% CI.

**Figure 9 microorganisms-11-02018-f009:**
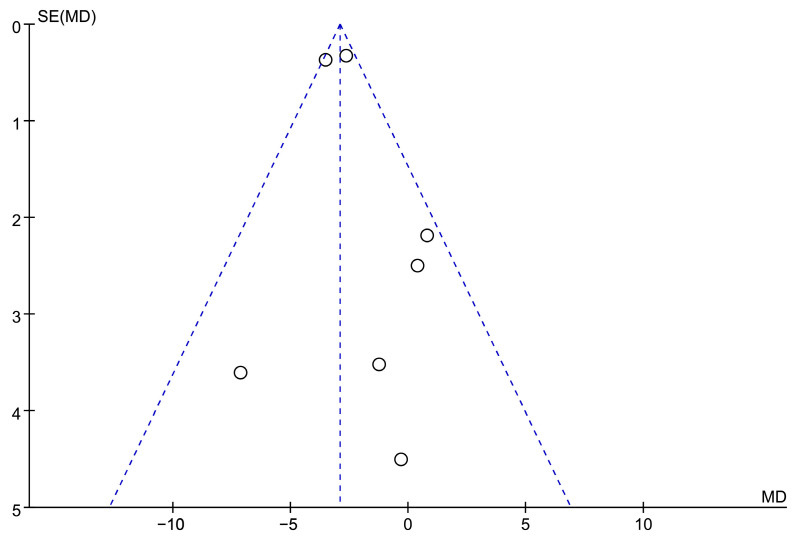
Funnel plot of included probiotics studies.

**Figure 10 microorganisms-11-02018-f010:**
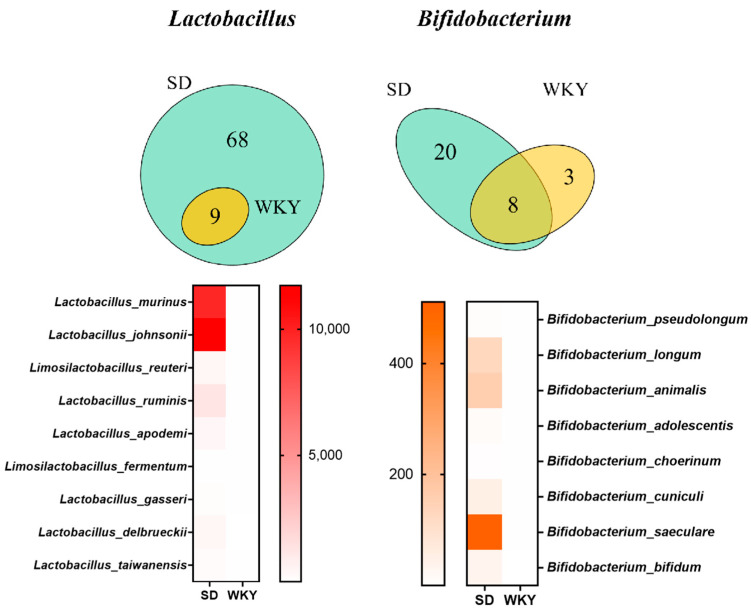
Venn and Heatmap revealed the probiotics content in the WKY rats and SD rats. The abundance and diversity of probiotics had decreased in WKY rats.

**Table 1 microorganisms-11-02018-t001:** Characteristics of the included studies and their interventions upon fatty acid administration.

Author Year	Country	Sample (Female)	Depression Criterion	Depression Assessments	Fatty Acid	Methodology	Ref.
Antypa, N. 2012	Netherlands	71 (58)	Beck Depression Inventory-II (BDI-II)	BDI-II	2.3 g of n-3 PUFA (including 1.74 g eicosapentaenoic acid (EPA) + 0.25 g docosahexaenoic acid (DHA) for 4 w	Double-blind; concealment allocation without statement; other anti-depressants with permission.	[27]
M Bot 2010	Netherlands	25 (13)	Composite International Diagnostic Interview	Montgomery Åsberg Depression Rating Scale (MADRS)	omega-3 (E-EPA) (1 g/day) for 12 w	Double-blind; concealment allocation without statement; other anti-depressants with permission.	[28]
Robert M Carney 2009	USA	122 (41)	Patient Health Questionnaire(PHQ-9)/BDI-II/Depression Interview and Structured Hamilton (DISH)	BDI-II/Hamilton Rating Scale for Depression (HAM-D)	930 mg of EPA and 750 mg of DHA for 10 w	Double-blind; other anti-depressants without permission.	[29]
Carney, R. M., 2019	USA	144 (56)	PHQ-9 and DISH	BDI-II/HAM-D/PHQ-9	50 mg/day of sertraline and 2 g/day of EPA for 10 w	Double-blind; without statement (permission) of concealment allocation/other anti-depressants	[30]
Jane Pei- Chen Chang, 2019	China	59 (38)	HAMD	BDI/HAMD	2 g per day of EPA and 1 g of DHA for 12 w	Double-blind; without statement (permission) concealment allocation is/other anti-depressants	[31]
Simin Dashti-Khavidaki, 2014	Iran	34 (17)	BDI	BDI	180 mg EPA and 120 mg DHA for 4 m	Double-blind; without statement (permission) of concealment allocation/ other anti-depressants	[32]
Frangou, S. 2006	British	51 (38)	HRSD	HRSD	1 g/day (*n* = 24) or 2 g/day (*n* = 25) of ethyl-EPA for 12 w	Double-blind; without statement (permission) of concealment allocation/other anti-depressants	[33]
Freeman, Marlene P. 2008	USA	51 (51)	Structured Clinical Interview for DSM-IV (SCID)	HAM-D and Edinburgh Postnatal Depression Scale (EPDS)	1.9 g/day EPA and DHA for 8 w	Double-blind; without statement (permission) of concealment allocation/other anti-depressants	[34]
Mary A. Fristad, 2016	USA	36 (21)	DSM-IV/Children’s Depression Rating Scale-Revised(CDRS-R)	CDRS-R	350 mg EPA 50 mg DHA, 68 mg other Ω3 (total 1870 mg) for 12 w	Double-blind; without statement(permission) of concealment allocation/other anti-depressants	[35]
Vilma Gabbay 2018	USA	51 (28)	Clinical Global Impressions—Improvement (CGI-I)	CDRS-R/BDI-II	Increased 0.6 g/d every 2 weeks (min:1.2 maximum possible dose of 3.6 g/d, combined EPA 2.4 g plus DHA 1.2 g	Double-Blind; without statement(permission) of concealment allocation/other anti-depressants	[36]
Leila Jahangard, 2018	Iran	50 (34)	DSM 5	BDI/MADS	1080 mg EPA, and 720 mg DHA per day for 12 w	Double-blind; without statement(permission) of concealment allocation is not stated/other anti-depressants	[37]
Seyed Ali Keshavarz, 2018	Iran	55 (55)	DSM-5	BDI	1080 mg EPA, and 720 mg DHA per day for 12 w	Double-blind, concealment allocation is not stated; other anti-depressants are not allowed.	[38]
Lesperance, Francois, 2011	Canada	432 (296)	Inventory of Depressive Symptomatology(IDS-SR30)	IDS-SR30 MADRS	1050 mg/d of EPA and 150 mg/d of DHA for 8 w	Double-blind; without statement(permission) of concealment allocation/other anti-depressants.	[39]
Lucas, M. 2009	USA	120 (120)	Psychological General Well-Being (PGWB)	20-item Hopkins Symptom Checklist Depression Scale (HSCL-D-20)/HAM-D-21	1.05 g E-EPA/d plus 0.15 g EHA/d for 8 w	Double-blind; without statement(permission) of concealment allocation/ other anti-depressants	[40]
Lauren B. Marangell, M.D. 2003	USA	35	DSM-IV	MADRS	2 g/day of DHA for 6 w.	Double-blind; without statement(permission) of concealment allocation is not stated/other anti-depressants	[41]
Mozaffari-Khosravi, Hassan, 2013	Iran	46 (26)	DSM-IV	HDRS	1 g/d of EPA or DHA for 12 w	Double-blind; without statement(permission) of concealment allocation/other anti-depressants	[42]
Mozurkewich, E. L. 2013	USA	80 (80)	BDI	BDI	EPA-rich fish oil supplementation (1060 mg EPA plus 274 mg DHA)	Double-blind; without statement(permission) of concealment allocation/other anti-depressants	[43]
Nemets, B. 2002	Israel	20 (17)	DSM-IV	HDRS	2 g/day E-EPA for 4 w	Double-blind; without statement(permission) of concealment allocation/other anti-depressants	[44]
Nishi, D. 2020	Japan and Taiwan	100 (100)	EPDS	HAMD	1800 mg omega-3 fatty acids (1206 mg EPA and 609 mg DHA) for 12 weeks	Double-blind; without a statement(permission) of concealment allocation, but permitted other anti-depressants	[45]
Nishi, D. 2016	Japan and Taiwan	31 (31)	EPDS	HAMD	1206 mg EPA and 609 mg DHA daily for 12 w	Double-blind; without statement concealment allocation; but permitted other anti-depressants.	[46]
Pomponi, M. 2014	Italy	24 (11)	DSM-IV	HDRS	800 mg/d DHA and 290 mg/d eicosapentaenoic acid for 6 m	Double-blind; without statement(permission) of concealment allocation is not stated/other anti-depressants	[47]
Rogers, Peter J. 2008	UK	190	DASS depression	DASS depression/BDI	630 mg EPA, 850 mg DHA, 870 mg olive oil for 12 w	Double-blind; without statement(permission) of concealment allocation/other anti-depressants	[48]
Rondanelli, Mariangela, 2010	Italy	46 (46)	Geriatric Depression Scale (GDS)	GDS	1.67 g of EPA and 0.83 g of DHA for 8 w	Double-blind; without statement(permission) of concealment allocation/other anti-depressants	[49]
Su, K. P. 2003	China	22 (18)	HRSD/ DSM-IV	HRSD	440 mg of EPA and 220 mg of DHA for 8 w	Double-blind; without statement(permission) of concealment allocation/other anti-depressants	[50]
Su, K. P., 2008	China	24 (24)	EPDS	HAMD	2.2 g of EPA and 1.2 g of DHA for 8 weeks	Double-blind; without statement(permission) of concealment allocation/other anti-depressants	[51]
Tajalizadekhoob, Yaser, 2011	Iran	66 (46)	Geriatric Depression Scale-15 (GDS-15)	GDS-15	180 mg eicosapentaenoic acid (EPA) and 120 mg DHA for 6 months	Double-blind; without statement(permission) of concealment allocation/other anti-depressants.	[52]

**Table 2 microorganisms-11-02018-t002:** Characteristics of included studies on probiotics pharmaceutical study.

Author Year	Country	Sample (Female)	Depression Criterion	Depression Criterion	Probiotics	Methodology	Ref.
Akkasheh, G. 2016	Iran	40 (34)	DSM-IV/HDRS	BDI	*Lactobacillus acidophilus* (2 × 109 CFU/g), *Lacticaseibacillus casei* (2 × 109 CFU/g), and *Bifidobacterium bifidum* (2 × 109 CFU/g) for 8 w	Double-blind; without statement (permission) of other antidepressant drugs	[23]
Arumugam, S. 2018	India	40 (34)	Diagnostic and Statistical Manual of Mental Disorders	HAM-D/MADRS/Center for Epidemiological Studies Depression Scale (CES-D)	*Bacillus coagulans MTCC 5856* (600 mg) for 90 d	Double-blind; without statement of other antidepressant drugs	[53]
Bahia Chahwan, 2019	Australia	71 (49)	BDI-II	BDI	2 g Ecologic^®^Barrier (2.5 × 109 CFU/g) is constituted of the following nine bacterial strains: *Bifidobacterium bifidum W23*, *Bifidobacterium lactis W51*, *Bifidobacterium lactis W52*, *L. acidophilus W37*, *Levilactobacillus brevis W63*, *Lacticaseibacillus casei W56*, *Ligilactobacillus salivarius W24*, *Lactococcus lactis W19* and *Lactococcus lactis W58* (total cell count 1 × 1010 CFU/day) for 8 w	Triple-blinded; without permission of other antidepressant drugs	[54]
Reininghaus, E. Z. 2020	Austria	61 (47)	HAMD/ BDI-II	HAMD/ BDI-II	*Bifidobacterium bifidum W23*, *Bifidobacterium lactis W51*, *Bifidobacterium lactis W52*, *Lactobacillus acidophilus W22*, *Lacticaseibacillus casei W56*, *Lacticaseibacillus paracasei W20*, *Lactiplantibacillus plantarum W62*, *Ligilactobacillus salivarius W24 and Lactococcus lactis W19* for 28 d	Double-Blind; with permission of other antidepressant drugs	[55]
Rita Baião 2021	UK	71 (45)	PHQ-9	PHQ-9	Four capsules. The probiotic (Bio-Kult^®^ Advanced, ADM Protexin Ltd.), consisted of 14 species of bacteria, (*Bacillus subtilis PXN*^®^ 21, *Bifidobacterium bifidum PXN*^®^ 23, *Bifidobacterium breve PXN*^®^ 25, *Bifidobacterium infantis PXN*^®^ 27, *B. longum PXN*^®^ 30, *Lactobacillus acidophilus PXN*^®^ 35, *Lactobacillus delbrueckii ssp. bulgaricus PXN*^®^ 39, *Lacticaseibacillus casei PXN*^®^ 37, *Lactiplantibacillus plantarum PXN*^®^ 47, *Lacticaseibacillus rhamnosus PXN*^®^ 54, *Lactobacillus helveticus PXN*^®^ 45, *Ligilactobacillus salivarius PXN*^®^ 57, *Lactococcus lactis ssp. lactis PXN*^®^ 63, *Streptococcus thermophilus PXN*^®^ 66), encapsulated at 2 × 109 CFU for 4 w	Double-blind, without permission of other antidepressant drugs	[56]
Romijn, A. R., 2017	New Zealand	79 (62)	Quick Inventory of Depressive Symptomatology (QIDS-SR16)/DASS-42	MADRS	*Lactobacillus helveticus R0052B. longum R0175* ≥ 3 × 109 CFU for 8 w	Double-blind; without permission of other antidepressant drugs	[57]
Zhang, X., 2021	China	69 (44)	DSM-5	BDI/HAMD	100 mL of a *Lacticaseibacillus casei* *Shirota* beverage (108 CFU/mL) for 9 w	Double-Blind; without permission of other antidepressant drugs	[58]

**Table 3 microorganisms-11-02018-t003:** Summary of the mechanism of probiotics on depression.

	Probiotics	Strain	Patients/Animal Models	Mechanism
1	*Bifidobacterium Longum*	*Bifidobacterium longum* (strain) 1714	Patients/Rat models	Mediating vagal tone/Increasing in brain-derived neurotrophic factor (BDNF)
2	*Lacticaseibacillus rhamnosus*		Rat models	Altering the neurotransmission of GABA
3	*Lactobacillus Helveticus*	*Lactobacillus helveticus* (strain NS8)	Rat models	Reducing blood pressure/Decreasing neuroinflammation
4	*Lactiplantibacillus plantarum*	*Lactiplantibacillus plantarum C29*	Rat models	Increasing BDNF/Decreasing inflammation
5	*Bifidobacterium Animalis*		Rat models	Inhibiting and/or reducing neuroinflammation
6	*Lacticaseibacillus casei*	*Lacticaseibacillus casei Shirota*/*Lacticaseibacillus casei W56*	Patients	
7	*Bifidobacterium Infantis*	-	Rat models	Modulating monoamine (serotonin and dopamine)/reducing inflammatory
8	*Bifidobacterium Breve*	*Bifidobacterium breve 1205/Levilactobacillus brevis W63*	Patients/Rat models	Unclear
9	*Lactobacillus Acidophilus*	*Lactobacillus acidophilus NCFM*	Rat models	Upregulating peripheral (and possibly central) CB2 receptors/modulating the cannabinoid and mu-opioid receptor
10	*Trans-Galactooligosaccharides*	-	Patients	Unclear

## Data Availability

The datasets used and/or analyzed during the current investigation are accessible upon reasonable requests from the corresponding author.

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
