# Peer review of "Current Insights into the Use of Probiotics and Fatty Acids in Alleviating Depression"

_microorganisms, 2023, doi:10.3390/microorganisms11082018_

Round 1
Reviewer 1 Report
With this review the Authors tried to sum up the more inclusive knowledge on probiotic and fatty acid efficicacy towards depression.
Despite the huge work made by the Authors, the article lack of attention to details
The title should be rephrased
The abstract should not contain abbreviations
Figure legends should be rephrases
Bacteria genus and species should be always written in italics
The main criticism of this work is represented by sentence structure, grammar, and language. I encourage the Authors to review the scientific English language used in the manuscript by an English mothertongue to enhance its contents. There are a number of sentences which do not make sense in addition to the use of wrong tenses. In a number of cases, this has led to misinterpretations of what the authors are trying to present.
The main criticism of this work is represented by sentence structure, grammar, and language. I encourage the Authors to review the scientific English language used in the manuscript by an English mothertongue to enhance its contents. There are a number of sentences which do not make sense in addition to the use of wrong tenses. In a number of cases, this has led to misinterpretations of what the authors are trying to present.
Author Response
- The title should be rephrased
Response: Thank you very much for your generous tips. Following your meaningful suggestion, we have revised the title of the article to better express the theme of our article. Please see page 1, line 2- 3 of the article.
- The abstract should not contain abbreviations
Response: Sincerely thank you for your constructive comments on the article. According to your kind suggestions, we have modified the abbreviations in the part of the abstract again. All modified parts have been marked in bright blue. Please see page 1, line 17-36 of the article. Thank you again.
3. Figure legends should be rephrases
Response: Sincerely thank you for your constructive comments on the inadequacies of the illustrations in this article. According to your suggestions, we have rephrased the illustrations in the article again. Once again, I would like to express my heartfelt thanks to you for your generous advice.
4. Bacteria genus and species should be always written in italics
Response: Sincerely thank you for your point out our limitation. According to your generous advice, we have revised all the Bacteria genus and species in italics . Due to the wide range of revisions, we can only mark them in bright blue. Sincerely thank you for your careful and serious suggestions.
5. The main criticism of this work is represented by sentence structure, grammar, and language. I encourage the Authors to review the scientific English language used in the manuscript by an English mothertongueto enhance its contents. There are a number of sentences which do not make sense in addition to the use of wrong tenses. In a number of cases, this has led to misinterpretations of what the authors are trying to present.
Response: Sincerely thank you for your impartial and generous comments on our article, as well as for pointing out the limitations of the article. According to your kind suggestions, we invited Dr. Xin Tao, from Department of Applied Linguistics of Capital Medical University, to undertake a thorough assessment of the paper, including spelling, grammar, and sentences, in order to make it more legible. Meanwhile, we have also attentively and carefully read the full text. The article's changed sections are all highlighted in bright blue. Thank you again for taking the time to read our investigation. We feel that the readability of the content has improved as a result of the comprehensive review. We also want to thank you from the bottom of our hearts for your help to us. All the modification in the article have been marked in bright blue.
Reviewer 2 Report
This manuscript describes a study designed to explore current evidence for probiotic and fatty acid therapy in depression and present a current option based on a meta-analysis of the literature and the data of metagenome DNA analysis from WKY rats.
Together, these findings suggest that fatty acids and probiotics have beneficial effects on depression, perhaps introducing a new strategy for treating depression.
However, several issues should be addressed before accepting the manuscript for publication.
1. Line 25: Currently, there is a lot of evidence that WKY rats are a good model of treatment-resistant depression (TRD), not MDD. The reason for such an interpretation is that in various models of depression these animals do not respond to standard antidepressant treatment. WKY rat displays behavioural and neurobiological phenotypes similar to that observed in resistance to common antidepressants. More importantly, this drug resistance in WKY rats can be overcome with TRD treatment i.e.ketamine or deep brain stimulation. Meta-analyses in this study refer to MDD, not TRD. Could the Authors comment on this issue?
2. Line 28: The Abstract should also contain a brief reference to the results, not the conclusion from the results only.
3. Line 38/39: Anhedonia is an inability to feel pleasure.
Social withdrawal is an equally important issue.
4. Line 51/52: Beck Depression Inventory is repeated.
5. Line 130: How many animals were sampled for metagenomic DNA analysis, female or male? Were the samples analyzed singly or in duplicate?
6. Please pay attention to the appearance of Table 1, Figures 3, and 5; standardize the fonts.
7. Line 215: Number of..? in depression rats. It would be better to use “WKY rats” than “depressed rats” in the ms. There are few models of depression in rats.
8. Line 221: weperformed
9. Line 240: In addition to the key mechanisms of depression in line 235, the Inflammatory theory of depression is equally recognized, which the Authors describe below.
10. Line 267: BBB - does the abbreviation appear anywhere else in ms?
11. Line 275: Phrases of type “As we all know” are avoided.
12. Line 293: When considering the contribution of different strains of bacteria to depression, it is useful to present a simple table of strain-level relationships in depressed patients/animal models.
13. Line 319: Please refer to the literature.
14. Missing spaces in several places.
Author Response
REVIEW 2
This manuscript describes a study designed to explore current evidence for probiotic and fatty acid therapy in depression and present a current option based on a meta-analysis of the literature and the data of metagenome DNA analysis from WKY rats.
Together, these findings suggest that fatty acids and probiotics have beneficial effects on depression, perhaps introducing a new strategy for treating depression.
However, several issues should be addressed before accepting the manuscript for publication.
- Line 25: Currently, there is a lot of evidence that WKY rats are a good model of treatment-resistant depression (TRD), not MDD. The reason for such an interpretation is that in various models of depression these animals do not respond to standard antidepressant treatment. WKY rat displays behavioural and neurobiological phenotypes similar to that observed in resistance to common antidepressants. More importantly, this drug resistance in WKY rats can be overcome with TRD treatment i.e.ketamine or deep brain stimulation. Meta-analyses in this study refer to MDD, not TRD. Could the Authors comment on this issue?
Response: Thank you for your thoughtful inquiries. We noticed that major depressive disorder (MDD) is common, often under-treated, and a major cause of disability and mortality globally when we evaluated the literature on depression. Based on it, we decided to focus on MDD in this study. According to the reference, undertreated WKY rats are an excellent model depression, we want to reveal whether the gut microbiome of the WKY rats differs from that of SD rats, which causes the WKY rats’ depression. However, we did not distinguish the type of depression in the WKY rats. We have also made a thorough and detailed analysis of this result according to the references, which may be attributed to the following two aspects.First of all, this is a comparison of the intestinal flora of two different strains of experimental animals, so there should be some bias in the comparison;Secondly, it may be in the selection of experimental animals, some of which are not enough to lead to some differences, which may not be well reflected.We especially hope that we can continue to improve the content and results of this part in future experiments, so as to lay a solid foundation for future research.Thank you once more for your thoughtful advice.
- Line 28: The Abstract should also contain a brief reference to the results, not the conclusion from the results only.
Response: Sincerely thank you for your valuable comments. According to your generous suggestion, we have revised, supplemented and improved the results part of the summary part again. Please see on the Page1 Line28-33. Thank you again for your generous advice.
- Line 38/39: Anhedonia is an inability to feel pleasure.Social withdrawal is an equally important issue.
Response: Thank you for your generous hint. We have reiterated the essential word and gained a complete and detailed comprehension of its meaning as a result of your passionate recommendation. We agree that the term "Anhedonia" is a bit improper in this context, thus we have replaced it with the more acceptable term "pessimism". We also applied “social withdrawal” instead of “social behavior”. We have always felt that by replacing words, we can better communicate the exact meaning. Please refer to the Page 2 Lines 42-44. Thanks again.
- Line 51/52: Beck Depression Inventory is repeated.
Response: Thank you very much for your careful reading of our article and pointing out the mistake in the article. According to your constructive comments, we have deleted the repeated places in the article. Please see on the Page2 Line56.
- Line 130: How many animals were sampled for metagenomic DNA analysis, female or male? Were the samples analyzed singly or in duplicate?
Response: Thank you very much for your thoughtful inquiries.In our experiment, we analyzed data from sampled metagenomic DNA coming from male rats, which does not imply that we have a bias against female laboratory animals, because it has been shown in experiments and literature reports that the intestinal flora of female rats differs during different physiological cycles, so we avoided this difference. Our choice of laboratory animals is also consistent with the guidelines of the Committee on Laboratory Animal Ethics(AEEI-2016-079). We're experimenting with male rats. In this section of the investigation, we will analyze the metagenomic DNA data from seven male rats, including three SD rats, and four WKY rats. Each sample was extracted genomic DNA, and then genomic DNA was randomly fragmented for next analysis. Thank you again for your hearty tip.
- Please pay attention to the appearance of Table 1, Figures 3, and 5; standardize the fonts.
Response: Thanks for your generous remind. We have carefully examined the charts again. According to the instructions for authors, we changed the fonts into Palatino Linotype. However, the figures were obtained by the Reviewmanager 5.3, which can not change the font. Luckily, all the fonts meet the requirements of instructions for authors. Thank you again for your careful reading.
- Line 215: Number of..? in depression rats. It would be better to use “WKY rats” than “depressed rats” in the ms. There are few models of depression in rats.
Response: Sincerely thank you for reading attentively and pointing out the flaws in our article directly. We modified the confusing section of the article based on your great feedback. Seven male rats, including three SD rats, and four WKY rats, were collected for metagenomic DNA study. Please refer to the Page13 Line 225. Thank you for your thoughtful suggestions.
- Line 221: we performed
Response: Thank you, in particular, for your meticulous exposition of our manuscript's restriction. We modified it due to your helpful recommendations. Please refer to Page 13, Line 217. Thank you once again.
- Line 240: In addition to the key mechanisms of depression in line 235, the Inflammatory theory of depression is equally recognized, which the Authors describe below.
Response: Thank you for the kind reminder. We re-examined the relevant literature, and there is substantial evidence that there is a close relationship between depression and inflammation. According to your advice, we have additionally supplemented and described this aspect of the theory in the paper. We feel that this change will be of significant value in further understanding the incidence of depression and inflammation, as well as having crucial therapeutic implications. Please refer to Page 14, Line 244-247. We'd want to thank you again from the bottom of our hearts for your kind recommendations.
Inflammatory processes have been implicated in the depression pathogenesis. Inflammation is likely a critical disease modifier, increasing vulnerability to depression. It is now well documented that dysregulation of both the innate and adaptive immune systems occurs in depressed individuals, hampered by antidepressant responseser1, which have been proven to reduce inflammation, whereas higher levels of baseline inflammation predicts worse treatment effectiveness for the majority of therapies. Previous evidence has shown an increase in proinflammatory cytokines, such as TNFα and IL-6 , in people suffering from depression.
Higher rates of depression and exhaustion have been shown with a variety of disorders linked with immune system activation, including allergies, autoimmune diseases (Type 1 diabetes, multiple sclerosis, systemic lupus erythematosus, and rheumatoid arthritis), and infections2. Depression is also common among those who have rheumatoid arthritis (RA). Studies reported a 74% greater incidence of depression compared to controls, with a frequency as high as 73.2%, while a meta-analysis discovered that 16.8% of RA patients have it.
Immune-related ailments and situations where the immune system is stimulated in response to infections demonstrate the relationship between immunological activity and depression and fatigue. Changes in immunological marker levels have also been linked to antidepressant response and have proven to be useful in predicting treatment success. Higher IL-6 levels indicated worse treatment response in depressive individuals with bipolar illness receiving antidepressant sleep deprivation therapy, correlating with prior antidepressant research3.
Overall, there is considerable evidence that immune system modifications may be one of the mechanisms through which antidepressant medicines work. There is significant evidence for a relationship between inflammatory activity and depression4.
Refs
1.Beurel E, Toups M, Nemeroff CB. The Bidirectional Relationship of Depression and Inflammation: Double Trouble. Neuron.2020 ,107(2): 234–256.
2.Jiang M, Qin P, Yang X. Comorbidity between depression and asthma via immune-inflammatory pathways: a meta-analysis. J Affect Disord. 2014,166:22–9.
3.Benedetti F, Lucca A, Brambilla F, Colombo C, Smeraldi E. Interleukine-6 serum levels correlate with response to antidepressant sleep deprivation and sleep phase advance. Prog Neuropsychopharmacol Biol Psychiatry. 2002 ,26:1167–70.
- Soosova MS, Macejova Z, Zamboriova M, Dimunova L. Anxiety and depression in Slovak patients with rheumatoid arthritis. J Ment Health. 2017,26:21–7.
- Line 267: BBB - does the abbreviation appear anywhere else in ms?
Response: Sincerely thank you for your kind comments regarding our article's flaws. According to your suggestions, we have delete “BBB” , as your comments, to make the statement more correct. Thank you for your help.
- Line 275: Phrases of type “As we all know” are avoided.
Response: According to your kind suggestion, we have deleted the phrase. Thank you.
- Line 293: When considering the contribution of different strains of bacteria to depression, it is useful to present a simple table of strain-level relationships in depressed patients/animal models.
Response: Sincerely thank you for your constructive feedback. Because the influence of different strains of bacteria counts on depression is so significant, we created a table to describe its detailed information. So that we can see the importance of gut flora, particularly probiotics, in depression more clearly. Please see on Page 16, Line 354.
Table 1. Summary of the Mechanism of Probiotics on Depression
| 
 | Probiotics | Strain | Patients/animal models | Mechanism | 
| 1 | Bifidobacterium Longum | Bifidobacterium longum (strain) 1714 | Patients/rat models | Mediating vagal tone / Increasing in brain-derived neurotrophic factor (BDNF) | 
| 2 | Lactobacillus Rhamnosus | 
 | Rat models | Altering the neurotransmission of GABA | 
| 3 | Lactobacillus Helveticus | Lactobacillus helveticus (strain NS8) | Rat models | Reducing blood pressure/ Decreasing neuroinflammation | 
| 4 | Lactobacillus Plantarum | Lactobacillus plantarum C29 | Rat models | Increasing BDNF/Decreasing inflammation | 
| 5 | Bifidobacterium Animalis | 
 | Rat models | Inhibiting and/or reducing neuroinflammation | 
| 6 | Lactobacillus casei | Lactobacillus casei Shirota (LcS)/Lactobacillus casei W56 | Patients | 
 | 
| 7 | Bifidobacterium Infantis | 
 | Rat models | Modulating monoamine (serotonin and dopamine)/reducing inflammatory | 
| 8 | Bifidobacterium Breve | Bifidobacterium breve 1205 /Lactobacillus brevis W63 | Patients/animal models | Unclear | 
| 9 | Lactobacillus Acidophilus | Lactobacillus acidophilus NCFM | Rat models | Upregulating peripheral (and possibly central) CB2 receptors/modulating the cannabinoid and mu-opioid receptor | 
| 10 | Trans-Galactooligosaccharides | 
 | Patients | Unclear | 
- Line 319: Please refer to the literature.
Response: Thank you very much for your kind reminder. We apologize for the absence of literature support in the appropriate context process. Simultaneously, in response to your request, we have searched and checked the appropriate literature and information and have inserted the corresponding references into the article. Please refer to Page 16, Line 321. Probiotics have been shown to influence a psychological trait linked to depression vulnerability and have a suppressive effect on depression. A study found that α-diversity increased significantly in adult patients with depression, whereas bacterial β-diversity did not change significantly1. However, the opposite patterns were found in childhood depression2. Previous studies have reported that several key functional bacteria at different taxonomic levels are associated with depression. Of note, it is important to note that these benefits are strain-specific3.
Refs
- Jiang H, Ling Z, Zhang Y, Mao H, Ma Z, Yin Y, Wang W, Tang W, Tan Z, Shi J, Li L, Ruan B. Altered fecal microbiota composition in patients with major depressive disorder. Brain Behav Immun. 2015 ;48:186-94.
- Ling Z, Cheng Y, Chen F, Yan X, Liu X, Shao L, Jin G, Zhou D, Jiang G, Li H, Zhao L, Song Q. Changes in fecal microbiota composition and the cytokine expression profile in school-aged children with depression: A case-control study. Front Immunol. 2022;13:964910.
- Gao J, Zhao L, Cheng Y, Lei W, Wang Y, Liu X, Zheng N, Shao L, Chen X, Sun Y, Ling Z, Xu W. Probiotics for the treatment of depression and its comorbidities: A systemic review. Front Cell Infect Microbiol. 2023;13:1167116. 14. Missing spaces in several places.
Response:Thank you very much for your helpful and careful ideas, as well as for explicitly pointing out the flaws in our article. We have corrected these flaws in the article based on your feedback. Thank you once more for your assistance.
Round 2
Reviewer 1 Report
The Authors replied to almost all the criticisms retrieved in the previous revision.
Please:
rephrase Figure 2 legend
remove "Depicted the" in Figure 8 legend
Author Response
The Authors replied to almost all the criticisms retrieved in the previous revision.
Response: Sincerely thank you for your affirmation of our response to the editor and reviewers’ comments.
Please:
rephrase Figure 2 legend
Respond: Sincerely thank you for your thoughtful comments about our article's flaws. We changed and enhanced the expression approach based on your ideas. Please see Line 173 - 175 on Page 9. Thanks again.
remove "Depicted the" in Figure 8 legend
Respond: Thank you very much for your kind advice. We removed the word "Depicted" based on your suggestions. Please refer to Page 14 Line 213. Thanks again.
